# Directionality of the injected current targeting the P20/N20 source determines the efficacy of 140 Hz transcranial alternating current stimulation (tACS)-induced aftereffects in the somatosensory cortex

Mohd Faizal Mohd Zulkifly[1,2,3]*, Albert Lehr[1], Daniel van de Velden[1], Asad Khan[4], Niels K. Focke[1], Carsten H. Wolters[4,5], Walter Paulus[1,6]

1 Department of Clinical Neurophysiology, University Medical Center, Georg-August University, Goettingen, Germany, 2 Department of Neurosciences, School of Medical Sciences, Universiti Sains Malaysia Health Campus, Kubang Kerian, Kota Bharu, Kelantan, Malaysia, 3 Brain and Behaviour Cluster, School of Medical Sciences, Universiti Sains Malaysia Health Campus, Kubang Kerian, Kota Bharu, Kelantan, Malaysia, 4 Institute for Biomagnetism and Biosignalanalysis, University of Muenster, Muenster, Germany, 5 Otto Creutzfeldt Center for Cognitive Neuroscience, University of Muenster, Muenster, Germany, 6 Department of Neurology, Ludwig Maximilians University, Munich, Marchioninistr, München

* faizal.zulkifly@usm.my

## Abstract

Interindividual anatomical differences in the human cortex can lead to suboptimal current directions and may result in response variability of transcranial electrical stimulation methods. These differences in brain anatomy require individualized electrode stimulation montages to induce an optimal current density in the targeted area of each individual subject. We aimed to explore the possible modulatory effects of 140 Hz transcranial alternating current stimulation (tACS) on the somatosensory cortex using personalized multi-electrode stimulation montages. In two randomized experiments using either tactile finger or median nerve stimulation, we measured by evoked potentials the plasticity aftereffects and oscillatory power changes after 140 Hz tACS at 1.0 mA as compared to sham stimulation (n = 17, male = 9). We found a decrease in the power of oscillatory mu-rhythms during and immediately after tactile discrimination tasks, indicating an engagement of the somatosensory system during stimulus encoding. On a group level both the oscillatory power and the evoked potential amplitudes were not modulated by tACS neither after tactile finger stimulation nor after median nerve stimulation as compared to sham stimulation. On an individual level we could however demonstrate that lower angular difference (i.e., differences between the injected current vector in the target region and the source orientation vector) is associated with significantly higher changes in both P20/N20 and N30/P30 source activities. Our findings suggest that the higher the directionality of the injected current correlates to the dipole orientation the greater the tACS-induced aftereffects are.

**Data Availability Statement:** All relevant data are within the paper and its Supporting information files.

**Funding:** MFMZ was supported by the Ministry of Education (MOE), Malaysia. This work was partly supported by the Göttingen Graduate Center for Neurosciences, Biophysics, and Molecular Biosciences (GGNB) of the Georg-August-Universität Göttingen. AK and CHW were supported by the priority program SPP1665 of the German Research Foundation (DFG), project WO1425/5-2, and by DFG project WO1425/10-1. The funders had no role in study design, data collection and analysis, decision to publish, or preparation of the manuscript.

**Competing interests:** The authors have declared that no competing interests exist.

## Introduction

Transcranial electrical stimulation (tES) techniques allow to alter neuronal excitability by guiding low-intensity currents through the brain. Transcranial direct current stimulation (tDCS) exhibits its effects by changes in cortical excitability via alterations of neuronal resting membrane polarization, and its excitatory or inhibitory aftereffects are depending on polarity [1,2]. Transcranial alternating current stimulation (tACS) which is a non-fixed polarity protocol can also modulate cortical plasticity depending on the stimulation frequency [3–7]. A stimulation at the 'ripple' frequency of 140 Hz induces an excitability increase when applied at 1 mA, very similar to tDCS at 1 mA [8]. At a lower intensity of 0.4 mA an excitability decrease was demonstrated [9]. We have chosen tACS at 140 Hz and 1 mA here since it avoids the polarization seen with tDCS. Thus, we could concentrate on showing the importance of direction alignment of the injected current at the targeted area with the dipole of the primary somatosensory evoked potential (SEP) component and simultaneously excluding that the induced effects might be due to polarity differences.

In this study we focus on the somatosensory cortex because of its comparatively simple dipole representations during stimulation. The P20/N20 source activity is located in Broadman area 3b which represents fingers and hand body surface [10–13]. tACS on the somatosensory cortex targeted to be aligned with the P20/N20 component might induce sensation related-effects as shown previously with tACS at alpha (10–14 Hz), beta (16–20 Hz) and high gamma (52–70 Hz) frequencies eliciting tactile sensations in the contralateral hand [14]. Also, tACS targeting the somatosensory cortex at the endogenous alpha-band activity decreases the functional connectivity of the somatosensory network [15]. Beyond neurophysiological effects, tACS also modulates behavioural outcomes such as cognitive performances [16–19] and perception [20]. We used tactile discrimination tasks to measure behaviour changes after stimulating a targeted P20/N20 source activity as its performance represents different levels of cognitive processing such as perception, recognition, working memory and decision making.

Many factors can affect tES outcome and accordingly may contribute to its variability [21–24]. These include brain states, brain anatomy, the induced electric fields in the brain and technical factors such as stimulation parameters and the stimulation montage. Most tES studies use a predetermined fixed stimulation montage to all participants neglecting individual anatomy differences potentially leading to suboptimal electric field distribution [25,26]. Other groups optimized the stimulation montage based on the electroencephalography (EEG) spectral power difference on the scalp maps topography between patients and a control group [27]. Another group optimized the stimulation electrode montage based on individually simulated electric fields in multiple compartment finite-element method (FEM)- head models which also addresses the fact that the optimized electric fields are limited by individual anatomical differences [28,29]. More attention to this anatomical variation allows to control variability by optimizing the current density in the target region, leading to improvement in behavioural outcomes [30]. Inter-individual differences in brain morphology such as cortical folding affect the optimal direction of induced current [31]. This addresses the importance of individually optimizing the stimulation electrode montage in order to reduce one source of variability which impairs tES efficacy [32].

In this study, we therefore took into account individual head and brain geometry in order to clarify further the importance of optimal current flow direction. We calculated the electric field strength to personalize the stimulation montage based on individual MRI data [33]. We specified the underlying source of the maximal P20/N20 component as the target for stimulation from EEG source reconstruction which was localized in the Brodmann area 3b in the somatosensory cortex [34], representing fingers and other hand areas [35]. We assumed that

the source orientation guides the optimal direction of current flow [36–38]. Accordingly, in the motor cortex, a previous study showed that motor evoked potentials (MEPs) were reduced in the orthogonal (perpendicular) but not in the parallel montage to the gyrus [37].

This reconstructed source was then used to determine the location of target and return electrodes. Variation of the position of the return electrodes alters e.g. the activity in the motor system [39]. Increases in electrode distance induce more distributed currents and increase the depth and magnitude of the electric field but they also reduce the shunting effects between electrodes [40]. Personalized stimulation electrodes (i.e., electrode placement tailored to the individually folded cortical patches) that employ a structural MRI of each subject were shown to induce a higher and larger spread of electric field distribution [41]. tACS at 20 Hz with a personalized electrode positioning induced an increase in the MEP amplitudes possibly overcoming the response variability issues [42]. Here we personalize stimulation montage and by this current direction. Overall, the key aim of the present study was to explore the dependency of modulatory effects of 140 Hz tACS on the somatosensory cortex on the tACS current flow direction. In addition, we looked for a possible relationship between plasticity aftereffects as measured by somatosensory evoked potentials and differences between the injected current vector at the target side, and the source orientation vector.

## Methods

### Participants

Seventeen healthy participants (9 male; age 24.00 ± 2.83 years (mean ± SD); range 20–30) were recruited in this study after giving written consent. They were right-handed as assessed by the Edinburgh handedness inventory [43]. Our participants had no history of neurological and psychiatric illnesses, no contraindication to brain stimulation and magnetic resonance imaging (MRI) and not the active smokers. In each session, the participant's history of medication use, alcohol, smoking and caffeinated products use was recorded because these factors may modulate cortical excitability and plasticity, brain oscillations and connectivity [23,44–49]. Participants were advised to refrain from caffeine, smoking and alcohol use a day before experiment and on the experimental days. As hormonal changes can be confounding factors to brain stimulation responses [50,51], sessions with female participants were conducted at least five days after the menses ended. One participant dropped out from session 5 and 6 due to a personal reason and the data analysis of that participant was carried out on the completed sessions. This study was approved by the Ethical Committee of the University Medical Center Göttingen and participants informed consent was conducted in accordance with the Declaration of Helsinki.

### Experimental design and paradigm

The experiments consisted of two parts. In a pre-experiment, EEG and an individual MRI scan were recorded to source localize the P20/N20 target elicited by median nerve stimulation. Then, the main experiment consisted of different sessions of EEG recordings before and after tACS or sham. We defined P20 as the maximum positive voltage value over the frontal pole. N20 is the maximum negative voltage value over the occipital pole which peaks between 18 to 22 ms after median nerve stimulation. N30 is defined as the maximum negative voltage value over the frontal pole and P30 is the maximum positive voltage value over the occipital pole which peaks between 28 to 34 ms after median nerve stimulation. In this paper, we use P20/N20 and N30/P30 to represent the components of both poles.

**Pre-experiment.** In this session, the localization of the P20/N20 source was performed after stimulating the right median nerve (Fig 1a).

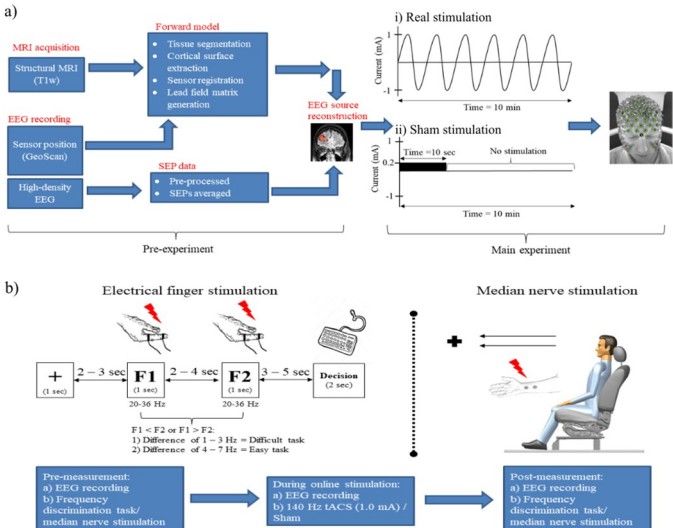

**Fig 1. Experimental design.** (a) This study consisted of two phases of measurements. In the pre-experiment, the stimulation target was determined from two separate sessions of MRI data acquisition and EEG recording. Both anatomical (MRI) and functional (EEG) data were subjected to the source reconstruction to obtain the information on the P20/N20 source dipole. (b) In the main experiment, both real and sham transcranial stimulation were carried out for 10 minutes on separate days using an individualized stimulation electrode montage which was calculated in (a). Measurements of the main experiment involved either the electrical finger stimulation or the median nerve stimulation in separate sessions. The order of the sessions was randomized with at least five days between each session. In the sessions with electrical finger stimulation, participants performed the frequency discrimination tasks. They were requested to discriminate which out of two frequencies was faster (F1 or F2). There were 70 trials in each task, and the order of the frequency differences was pseudo-randomized. Out read performance was grouped as an easy task if the frequency difference was 4–7 Hz, whereas a difficult task was scaled with 1–3 Hz difference. Each session started with pre-measurement, followed by 10 minutes of transcranial electrical stimulation and the post-measurement. Abbreviations: MRI = magnetic resonance imaging; EEG = electroencephalography; SEP = somatosensory evoked potential; tACS = transcranial alternating current stimulation; F1 = frequency 1; F2 = frequency 2.

In session 1, the EEG recording with a 256-channel Geodesic Sensor Net (Electrical Geodesics, Inc.) was performed during electrical stimulation delivered to the right median nerve at the wrist via two stimulation electrodes, the cathode allocated proximately by 2 cm to the anode. Non-painful stimuli to the median nerve were applied at the motor threshold. The electrical stimuli were generated by the isolated constant current stimulator (DS5, Digitimer, UK). Rectangular pulses with a duration of 0.2 ms and a frequency of 3 Hz were applied for 10 minutes [52]. After completing the EEG recording, GeoScan was used to digitize the electrode sensor positions (EGI, Eugene, OR, USA) [53]. In session 2, a structural MRI was obtained with a Magnetom PRISMA 3T scanner (Siemens™) using a 32-channel phased-array head coil. T1-weighted whole-brain anatomical scans were acquired with the 3D turbo fast low angle shot sequence (repetition time: 2250 ms, echo time: 3.25 ms, inversion time: 900 ms, flip angle: 9 deg, isotropic resolution: 1 mm³).

**Main experiment.** In the next sessions, either an electrical right finger stimulation or a right median nerve stimulation was performed in separate sessions in a randomized order (Fig 1b). There were three phases of measurements. Simultaneous EEG recording and either finger or median nerve stimulation were performed first in the pre-measurement and then during either tACS or placebo stimulation in a randomized order (see "Transcranial Alternating Current Stimulation (tACS)" section for a detailed description). Finally, post-stimulation measurement was repeated identically to the pre-measurement.

*Electrical finger stimulation (session 3 and 4)*. Stimulation was delivered to the right index finger, and stimuli were generated by the isolated constant current stimulator (DS5, Digitimer, UK). Digital ring electrodes (Digitimer, UK) were applied on the right index finger with the cathode to the distal and anode to the proximal phalanx. A conductive gel was applied to the electrodes (Signa gel®, Parker Laboratories, Inc., USA). Individual stimulation intensity was determined at the beginning of every session and was adjusted to 2.5 times the sensory threshold. During the frequency discrimination task, two different stimulation frequencies were applied (see details below), and participants had to determine which frequency was higher. A trial run was carried out before the real test to familiarize the participants with the tasks. No feedback was given in the real test. In the beginning, a pre-measurement was carried out while the electrical finger stimulation was applied simultaneously with EEG recordings. Participants were seated in a comfortable chair in front of a computer. The right hand of the participants was placed under the table, and the left hand was used to give a response via a keyboard. They were asked to reduce muscle movements and eye blinks during this measurement. Next, during online stimulation measurement, participants were stimulated transcranially with tACS or sham. Participants were awake, and EEG was recorded during transcranial stimulation. No electrical finger stimulation was carried out during online tACS or sham measurement. Post-measurement effects were measured identically to the pre-measurements.

*Frequency discrimination task*. The two-alternative frequency discrimination task we used was based on a previous study in monkeys [54] and in humans [55]. The Signal software (Cambridge Electronic Design Ltd., Cambridge, UK) was used to control the stimulation sequence and to synchronize it with EEG recordings, the electrical finger stimulation and the visual cues (e.g. fixation, F1, F2 and Decision; see Fig 1b) which was generated with PsychoPy software [56]. In each trial, the right index finger was stimulated with a first frequency (F1) and a second frequency (F2) for 1 sec (Fig 1b). The trial started with the subjects being asked to fixate for 1 sec a fixation cross followed by F1 after a 2–3 sec interval (randomly jittered in steps of 1 sec). F2 was applied after another 2–4 sec interval (randomly jittered in steps of 1 sec). Participants were required to respond after the "Decision" cue appeared on the computer screen after a 3–5 sec interval (randomly jittered in steps of 1 sec) with a keyboard-click using the left hand within a 2 sec interval either pressing key "1" if F1 frequency was higher than F2 or key "2" if F1 frequency was lower than F2. The range of both frequencies was between 20 and 36 Hz and the absolute difference between F1 and F2 for each trial was 1–7 Hz. Participants were allowed to blink during a fixation period and were advised to reduce eye blinks and muscle movements during the task to obtain trials without contamination with eye blinks and muscle artefacts in the EEG data.

*Median nerve stimulation (session 5 and 6)*. Participants sat upright on a comfortable chair. Rectangular pulses with a duration of 0.2 ms and a frequency of 1 Hz were applied for 18 minutes. During this measurement, the participants were asked to fix their eyes on the fixation cross in front of them and to reduce muscle movement and eye blink artefacts as much as possible. In the next phase this was repeated with online tACS or sham but without median nerve stimulation for 10 minutes. After that, post-stimulation measurement was repeated as in the pre-measurement.

## GTEN planning

In order to target the P20/N20 source we followed a previously published GTEN planning protocol [57]. We defined the first peak at the latency of 20 ms and localized the underlying source activity in the somatosensory cortex as P20/N20 individual target source. First, we identified the individual target via sLORETA source analysis of the P20/N20 component, followed by

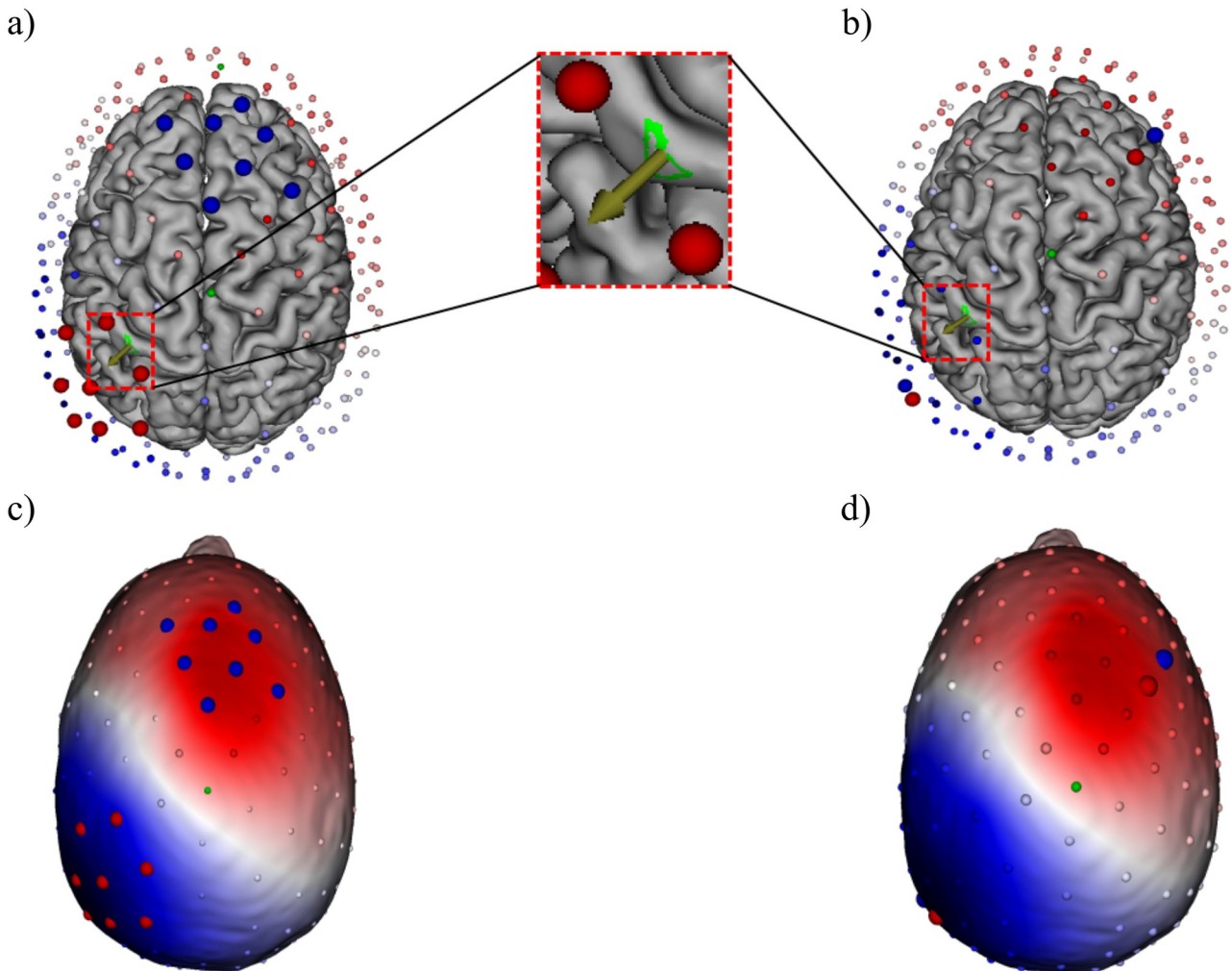

**Fig 2. Stimulation montage.** An example of the stimulation electrodes used for (a) tACS stimulation and (b) for sham stimulation. The electrode positions were optimized to best target the sLORETA localized P20/N20 source activity peak, which is shown by the arrow in the green area. (c) Optimized stimulation electrodes are located over the two poles (peak and trough) of the P20/N20 component. Blue indicates the negative voltage pole (N20) and red the positive voltage pole (P20). Eight circles in red are target electrodes of one polarity and another eight circles in blue of the opposite polarity. (d) Two electrodes were used for sham stimulation in each pole.

thresholding to the peak source activity, resulting in the dipolar P20/N20 reconstruction as indicated by the arrow in the green area in Fig 2a and 2b. We targeted the voxel with the highest sLORETA source activity in the region of interest (ROI) as the main target to generate a stimulation montage (refer to Fig 2a, a green area shown as ROI). Then, we specified the total current stimulation intensity for a least number of target electrodes (red) and return electrodes (blue) to inject the specified intensity based on a maximum current per electrode constraint. After that, the stimulation montage and locations of the electrodes were automatically generated by the GTEN module based on the reconstructed P20/N20 target source. On average, there were 12–20 electrodes (range: 6–10 electrodes per each cluster) used in the tACS condition (Fig 2c); four electrodes (2 electrodes per each cluster) were used for the sham condition (Fig 2d).

## Transcranial Alternating Current Stimulation (tACS)

Transcranial AC stimulation with a 140 Hz sinusoidal waveform was delivered by using the GTEN 100 system (EGI, Eugene, OR, USA) with the 256-channel HydroCel Geodesic Sensor Net (HCGSN) through Ag/AgCl electrodes. The target electrodes above the somatosensory cortex (S1) (average electrode: 7, range: 6–10 electrodes) were placed over the negative voltage potential cluster above the left S1, and the return electrodes (average electrode: 7, range: 6–10 electrodes) were at another cluster over the positive voltage potential (Fig 2c). These electrode positions were generated by the GTEN Planning Module as described above, and each position varied from one participant to another depending on their P20/N20 dipole position and orientation. Electrode size was 1 cm$^2$. The maximum current for each electrode was less than 0.2 mA due to device intensity constraints. We used a mixture of Elefix conductive paste (Nihon Kohden, Tokyo, Japan) and lidocaine cream 40 mg/g (GALENpharma GmbH, Kiel, Germany) as the conductive material between the electrode and scalp. The tACS stimulation duration was ten minutes with a current intensity of 1 mA following the original protocol of Moliadze and colleagues who showed intensity dependent excitatory or inhibitory after-effects in the motor cortex [8,9]. In the sham stimulation, we used a similar montage from a previous study with a minor modification of the stimulation parameters [57]. There were four electrodes used for sham stimulation, two of each of the active and return electrode clusters (Fig 2d). To achieve this, we chose the electrode with the lowest current level from the target electrode cluster of the tACS stimulation montage and manually set it as an anode with 0.1 mA current intensity for the first 10 seconds. After that, we chose one more electrode in close proximity to set it as a cathode with -0.1 mA current intensity. We used the same steps to determine the stimulation electrodes for the return electrodes cluster. In the sham stimulation condition, we applied a pulse stimulation waveform at 0.5 Hz for 10 seconds (pulse width: 1 ms, interpulse interval: 1999 ms) with a total of 0.2 mA stimulation intensity (see Fig 1a for an overview of the pulse configuration). This sham stimulation montage was intended to induce a similar skin sensation as in tACS for the purpose of blinding. Initial stimulation electrode impedances were kept below 100 kΩ (target electrode impedance: 9.51 ± 5.48 kΩ (mean ± S.D), return electrode impedance: 11.84 ± 6.71 kΩ (mean ± S.D)). The participants filled out questionnaires regarding stimulation-related sensations after each session.

## Data acquisition and analysis

EEG was recorded using the 256-channel Geodesic Sensor Net (Electrical Geodesics, Inc.). Signals were digitized at a sampling rate of 1000 Hz. The electrode impedances from all channels were kept below 50 kΩ. We applied an offline line noise filter at 50 Hz and carried out bad channel replacement using the Net Station tools (Electrical Geodesics, Inc.). Channels were labelled as bad when the system indicated a higher electrodes impedance of > 100 kΩ as a result of dried electrodes. Most of the rejected electrodes were on the face and neck, which did not influence signal analysis of our region of interest. EEG data were imported to EEGLAB for further pre-processing steps.

**Pre-experiment.** Pre-processed data were reimported to the Net Station tools (Electrical Geodesics, Inc.). As we intended to determine an individual's source localization and stimulation montage, an average SEPs waveform was obtained from trials averaging.

*Electroencephalography (EEG).* The signal software (Cambridge Electronic Design Ltd., Cambridge, UK) was used to control the timing synchronization between electrical stimulation on the median nerve with EEG recording. Pre-processing of EEG data was performed using custom scripts and EEGLAB 14.1.2 [58] within the Matlab environment version R2017a (Mathworks). We adapted the scripts and pre-processing pipelines following Stropahl and

colleagues [59]. An independent component analysis (ICA) was computed to attenuate physiological artefacts such as eye blinks, lateral eye movements and electrocardiogram artefacts. Manual artefacts rejection was carried out for each trial as well to remove remaining artefacts. After cleaning the continuous data, EEG data were filtered with a low-pass FIR filter (cut-off frequency of 250 Hz) and a high-pass FIR filter (cut-off frequency of 30 Hz). Data were referenced against a common average reference before being segmented relative to the stimulus onset into 700 ms epochs (200 ms pre-stimulus and 500 ms post-stimulus). After that they were subjected to baseline correction to the 200 ms pre-stimulus time window. The method of joint probability was used to remove the epochs with extreme artefacts (threshold of 4 standard deviations) [59]. Preprocessed data were exported to EGI for source reconstruction and GTEN (Geodesic Transcranial Electrical Neuromodulation) stimulation planning.

*Individualized head model*. The Modal Image Pipeline (EGI, Eugene, OR, USA) was used to segment MRI data into seven tissue types: eyeball, flesh, skull, cerebrospinal fluid (CSF), grey matter (GM), white matter (WM) and air. An atlas template of computed tomography (CT) (1 mm x 1 mm x 1mm) was non-linearly warped to the participant's MRI tissues. A detailed description of the tissue segmentation and the CT warping procedure can be found in [60]. Then, the cortical surface was parcelled into patches using the triangular meshes. We used 2400 dipole patches per hemisphere, and the size of each patch was ~ 1 cm$^2$. The perpendicular orientation informs the direction of current flow from the cortex to the scalp. It was computed by averaging the perpendicular directions of vertices within the patch. We then co-registered the electrode sensor positions to the scalp surface of the head model. Finally, a lead-field matrix (LFM) was computed using a finite difference method (FDM) [61]. We used the default conductivity values (in Siemens/meter) of each tissues; eyeball = 1.5, scalp = 0.44, skull = 0.018, CSF = 1.79, GM = 0.25, WM = 0.35 and air = 0.0 [57,62].

**Main-experiment.** EEG data pre-processing was carried out using EEGLAB 14.1.2 [58] as a signal processing toolbox. Later, pre-processed EEG data were analysed using the Fieldtrip [63] software with Matlab R2017a (The MathWorks, Inc., Natick, MA) as a platform. Further details of EEG and source analyses can be found below.

*Behavioural data analysis*. Task types were grouped into easy and difficult tasks. A frequency difference of 1 and 3 Hz was a difficult task type while a frequency difference of 4 and 7 Hz was grouped as an easy task type. We compared the differences in the frequency discrimination performance as represented by correct responses both in task types and also changes before and after the stimulation.

*Time-domain analysis*. Data from the median nerve stimulation sessions were down-sampled to 500 Hz. A continuous EEG data was filtered with a low-pass FIR filter (cut-off frequency 90 Hz) and a high-pass FIR filter (cut-off frequency 2 Hz) using *"pop_eegfiltnew"*. We removed 57 channels which were at the two last rows on the neck and channels on the face as they were noisy. This removal did not affect our region of interest. The EEG data were referenced against average reference before being segmented relative to the stimulus onset into 700 ms epochs (200 ms pre-stimulus and 500 ms post-stimulus). Data were then subjected to a baseline correction to the 200 ms pre-stimulus time window. Noisy epochs were removed using the *"pop_jointprob"* function (threshold of 4 standard deviations) [59]. As outlined in the pipelines by [59], we performed an ICA and the semi-automatic algorithm CORRMAP [64] to remove the components with eye blinks, eye movements and heart artefacts. A default correlation coefficient threshold (r $\geq$ 0.8) was used. Remaining artefacts were excluded by visual inspection.

Single-subject average waveforms were then averaged to obtain group-level average waveforms. We examined two early components of SEPs (i.e. P20/N20 and N30/P30). Visualization

in sensor space was performed over seven electrodes which were commonly used for more than half of the participants (n = 16).

*Spectral analysis.* A continuous raw EEG data set from the electrical finger stimulation sessions was down-sampled to 500 Hz, and high-frequency noise was removed by applying a low pass FIR filter (cut-off frequency 160 Hz) and a high pass FIR filter (cut-off frequency 1 Hz) using *"pop_eegfiltnew"*. Data were re-referenced against a common average reference and were epoched separately according to F1 and F2. Both segments were 4 sec long (1-sec pre-stimulus and 3-sec post-stimulus). Baseline correction was carried out against the pre-stimulus time window. Noisy epochs were removed using the *"pop_jointprob"* function (threshold of 4 standard deviations) [59]. An ICA and the semi-automatic algorithm CORRMAP, the EEGLAB plug-in [64] were carried out on the epoched data to remove the eye blinks, eye movements and heart components with a default correlation coefficient threshold ($r \geq 0.8$). Data were again subjected to a visual inspection to remove any remaining artefacts.

To explore the modulatory effects of 140 Hz tACS on evoked and induced oscillatory power in the somatosensory cortex we conducted a spectral analysis using Fieldtrip [63] and Matlab (The MathWorks, Inc., Natick, MA). Time-frequency (TF) spectral power analysis between 6 and 45 Hz was obtained by applying a tapered sliding window convolution using a single Hanning taper and an adaptive time window of five cycle length. Evoked power was computed for each stimulus condition by applying the TF transformation to the average waveform. This power indicates a phase-locked activity. For analysis of induced power, the average waveform was subtracted from the waveform of each trial before applying the TF spectral analysis to the single-trial data. Thus, the averaged TF spectra resulted in an estimate of purely induced (non-phase-locked) oscillatory power. For visualization, changes in spectral power over time were expressed as power changes relative to a pre-stimulus baseline period (500–200 ms before stimulus onset) in each respective frequency. Relative induced power decreases over time are known as event-related desynchronization (ERD) and relative induced power increases are known as event-related synchronization (ERS) [65].

*Source analysis.* The sources of evoked potential and oscillatory EEG power were reconstructed in Fieldtrip [63]. For each participant, a forward model was constructed. A three-layer-boundary element method (BEM) head model consisting of scalp, skull and brain compartments was extracted from the subject-specific MRI. The electrode alignment was initialized using subject-specific fiducial markers (i.e. nasion, right and left periauricular points). After that, the electrode positions were warped to the scalp surface of the head model. To achieve a comparable source model for the group, a volumetric template grid with a grid space distance of 5 mm was constructed based on the source model template. This volumetric template grid was then non-linearly warped to each subject-specific head model. The leadfield matrix that describes each sensor sensitivity of each voxel in the source model was then computed.

## Statistical analysis

Statistical analyses were performed using the Statistical Package for the Social Science (IBM SPSS statistics 26; IBM Corp., Armonk, NY, USA) and Fieldtrip [63]. In SPSS, the Shapiro-Wilk test was used to test the data distribution for normality. Non-parametric tests were used if the normality assumption was violated. A repeated measures (rm) ANOVA was used to examine the effects of stimulation on the discrimination performance with time and stimulation as within-subject factors ($\text{TIME}_{2 \text{ levels}}$ x $\text{STIMULATION}_{2 \text{ levels}}$) and task types as between-subject factor. TIME refers to pre- and post-stimulation and STIMULATION refers to the stimulation type (i.e., tACS and Sham). A cluster-based non-parametric permutation test [66]

was used to compare the SEPs amplitude changes before and after stimulation with sham and tACS. For oscillatory power analysis, a cluster-based non-parametric test was used to compare the oscillatory evoked and induced power changes before and after stimulation with sham and tACS. Randomization tests on the dependent sample with 1000 re-samplings (two-tailed) were conducted on the data.

We started with an exploratory analysis to examine the relationship between the vectors connecting the average anodes position with the average cathodes position relative to the P20/N20 reconstructed source. Specifically for this analysis, we used equivalent current dipoles (ECDs) to fit the P20/N20 topographies at 20 ms. This source reconstruction method was the preferred option to localize the mainly tangentially oriented sources underlying the P20/N20 components [13,67,68]. An individual dipole was reconstructed for each participant using a subject-specific head model which was generated in Fieldtrip as mentioned in "Source analysis section" above. For each individual, we averaged their target stimulation electrodes to obtain an average point of all target electrodes. Similarly, we obtained an average point of all return electrodes by averaging the return electrodes. This so-called *electrode vector* between the two average points, considered to approximate the *injected current vector at the target side*, was used to estimate the angular difference to the *P20/N20 source orientation vector* by taking the scalar product between both vectors. A scalar product of zero indicates that the electrode vector was perpendicular to the source orientation vector whereas a scalar product of one indicates parallelism of electrode and source orientation vectors. The absolute difference of P20/N20 amplitude, N30/P30 amplitude, P20/N30 complex, P20/N20 and N30/P30 source activities were correlated with the above electrode-source scalar product, assuming that a scalar product of one generates larger effects than a scalar product of zero [36–38]. Our procedure is motivated by Helmholtz' reciprocity principle, where it can be proven for a two-electrode-montage that a cathode at the P20 potential peak and an anode at the N20 potential trough leads to maximal intensity and parallelism at the target dipole [69]. Pearson or Spearman correlation was used to examine the relationship between the electrode-source scalar product and the SEP amplitudes and source activities. Data were reported as mean ± SD, unless otherwise specified and the statistical significance was set at p < 0.05 both in SPSS and in Fieldtrip.

## Results

### Stimulation perceived sensations

The stimulation-related sensation perception (itching, pain, burning, warming and tingling beneath the stimulation electrodes) in the tACS sessions was higher than the sensation reported in the sham sessions with 62.1% and 37.9% respectively ($\chi^2$ (1, $N = 66$) = 3.01, p = 0.08; see Supplementary Material, S1 Table).

### Effects of tACS on frequency discrimination performance

The average of the correct responses in the easy task (range: 81.8%–87.1%) was significantly higher compared to the difficult task (range: 63.3%–66.3%) both in the tACS and the sham condition ($F(1,32) = 36.31$, $p < 0.001$). There were no significant main effects of TIME ($F(1,32) = 3.77$, $p = 0.06$) and STIMULATION ($F(1, 32) = 0.50$, $p = 0.49$) on the task performances. An interaction of the factor TIME and STIMULATION on the performances was also not significant ($F(1, 32) = 0.43$, $p = 0.52$) (Fig 3). This indicates that the performance in the tactile tasks does not increase over time and the applied tACS on the somatosensory cortex does not significantly enhance the performance. (see Supplementary Material, S2 Table).

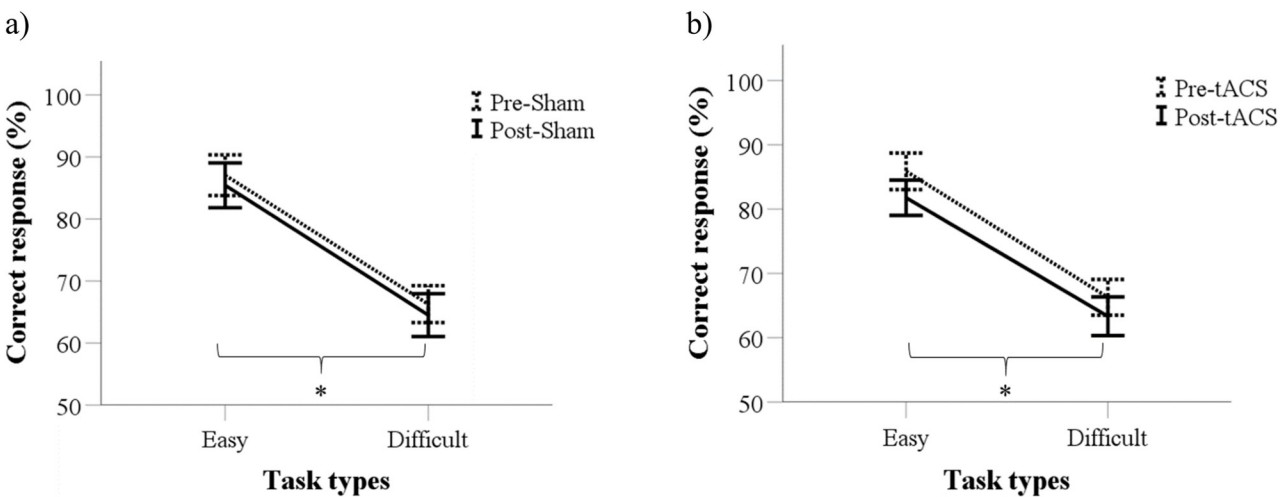

**Fig 3. Performance in the frequency discrimination tasks.** (a) Average performance in the Sham conditions based on task difficulty. (b) Average performance before and after tACS in two groups of task difficulty. * p < 0.05.

### Plasticity after-effects of tACS on the somatosensory cortex in the median nerve stimulation condition

There was no significant difference in the median nerve stimulation intensities between tACS condition (intensity 16.41 ± 3.31 mA (mean ± SD)) and sham condition (intensity 14.71 ± 3.08 mA (mean ± SD)). (see Supplementary Material, S3 Table). The early peaks of the P20/N20 and N30/P30 amplitudes in pre- and post-stimulation EEG both in sham and tACS conditions did not show significant group changes (Fig 4).

### tACS effects on evoked power changes during tactile discrimination tasks

There was no significant difference in the tactile stimulation intensities between tACS condition (intensity 2.57 ± 0.82 mA (mean ± SD)) and sham condition (intensity 2.65 ± 0.61 mA (mean ± SD)). (see Supplementary Material, S3 Table). There were no significant changes in the absolute power of the pre- and post-baseline time window in the sham group at any frequency (Fig 5a, top panel). However, the absolute EEG power of the pre-F1 stimulus time window was significantly lower compared to the power of the post-F1 stimulus at 10 Hz (cluster-based permutation test, p < 0.05; see Fig 5a, top panel). In the tACS group, the absolute power changes of the pre-baseline time window were significantly lower than the post-baseline time window at 7 Hz and 8 Hz (cluster-based permutation test, p < 0.05; see Fig 5a, bottom panel). There was no significant difference in the absolute power changes in the pre- and post-F1 stimulus time window at any frequency. (Fig 5a, bottom panel). Time-frequency analysis demonstrated a non-significantly increase of power at the stimulation electrodes at 5–12 Hz within 500 ms, and a non-significantly increase of power at 17–25 Hz within 200 ms during the stimulus F1 presentation before and after stimulation with sham and tACS. (Fig 5b and 5c). For stimulus F2, in the sham group, there was a non-significantly increase in the absolute power of the pre- and post-baseline stimulus and also no significant increase in the pre- and post-F2 time windows on the oscillatory activity at any frequency (Fig 5d, top panel). In the tACS group, there were no significant absolute power changes in the pre- and post-baseline time windows. The absolute power of the Pre-F2 stimulus time window was significantly lower compared to the Post-F2 stimulus time window at 9 Hz (cluster based permutation, p < 0.05;

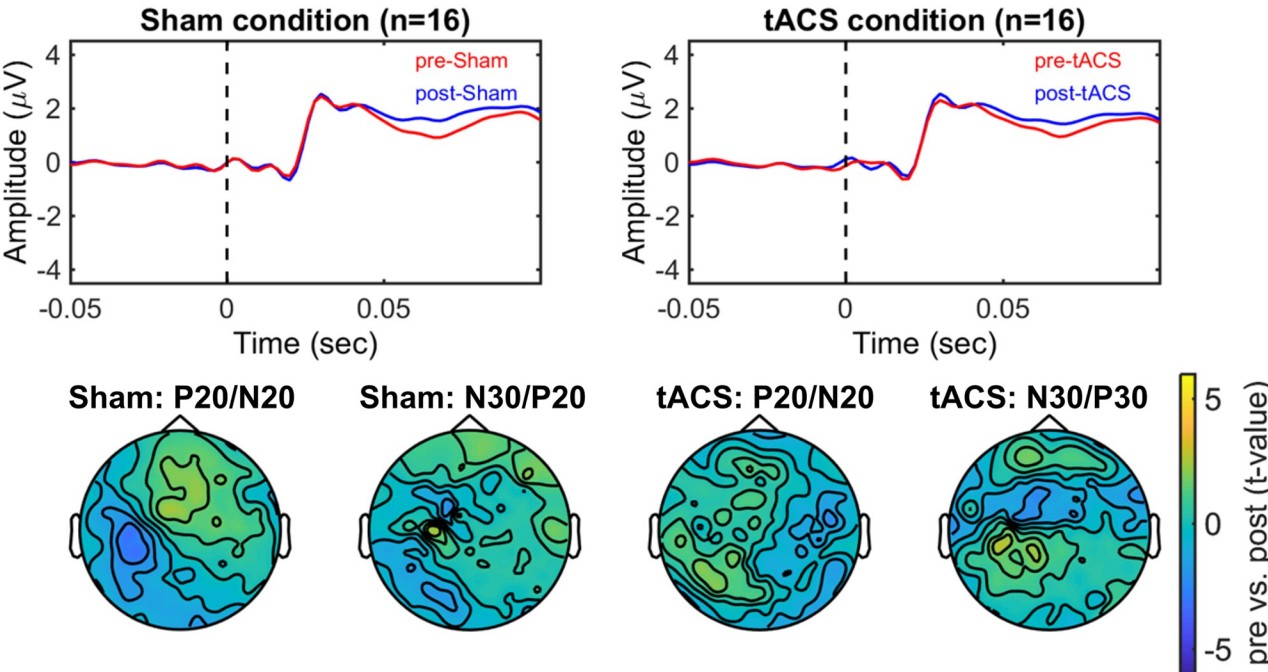

**Fig 4. Modulatory effects of tACS on the early components of the somatosensory evoked potentials (SEPs).** Top figures show the average amplitude evoked by median nerve stimulation measured before (pre-) and after (post-) stimulation with Sham and tACS over the somatosensory area (i.e. target electrodes). Bottom figures show the scalp topography of the differences between pre and post measurements with Sham and tACS for P20/N20 and N30/P30 amplitudes (Cluster-based permutation test, p > 0.05).

see Fig 5d, bottom panel). Evoked power at the stimulation electrodes was non-significantly higher at 5–12 Hz within 500 ms. Also, the evoked power was non-significantly higher at 17–25 Hz within 200 ms before and after stimulation with sham and tACS during stimulus F2 presentation. (Fig 5e and 5f).

## tACS effects on induced power changes during tactile discrimination tasks

Fig 6a illustrates event related desynchronization (ERD) changes during F1 stimulus presentation. Higher ERD of alpha activity occurred over the left somatosensory and bilateral posterior areas The ERD of beta activity was stronger in the left somatosensory area than in the right cortex (Fig 6b). Tactile stimulation induced a non-significant power decrease in the alpha (8–12 Hz) and also the beta band (17–25 Hz) after stimulation with sham and tACS. (Fig 6c). Similarly, there were no significant power changes during the F2 stimulus in the alpha and beta bands after stimulation with sham and tACS (Fig 6d–6f).

## Relationship of the angular difference between stimulation electrode and source orientation vectors with SEP amplitudes and source activities

The average angular difference between the stimulation electrode vector and the source orientation vector was 37.76 ± 19.82˚ (mean ± SD, range 0˚–65.80˚). There were significant relationships between the angular difference and the difference in P20/N20 and N30/P30 source activities in the tACS but not in the sham condition (Fig 7a–7d). Fig 7b shows that a larger angular difference is associated with smaller P20/N20 source activity in the tACS condition (r = -0.63, p < 0.05). A consistent finding is also shown in Fig 7d with a larger angular difference

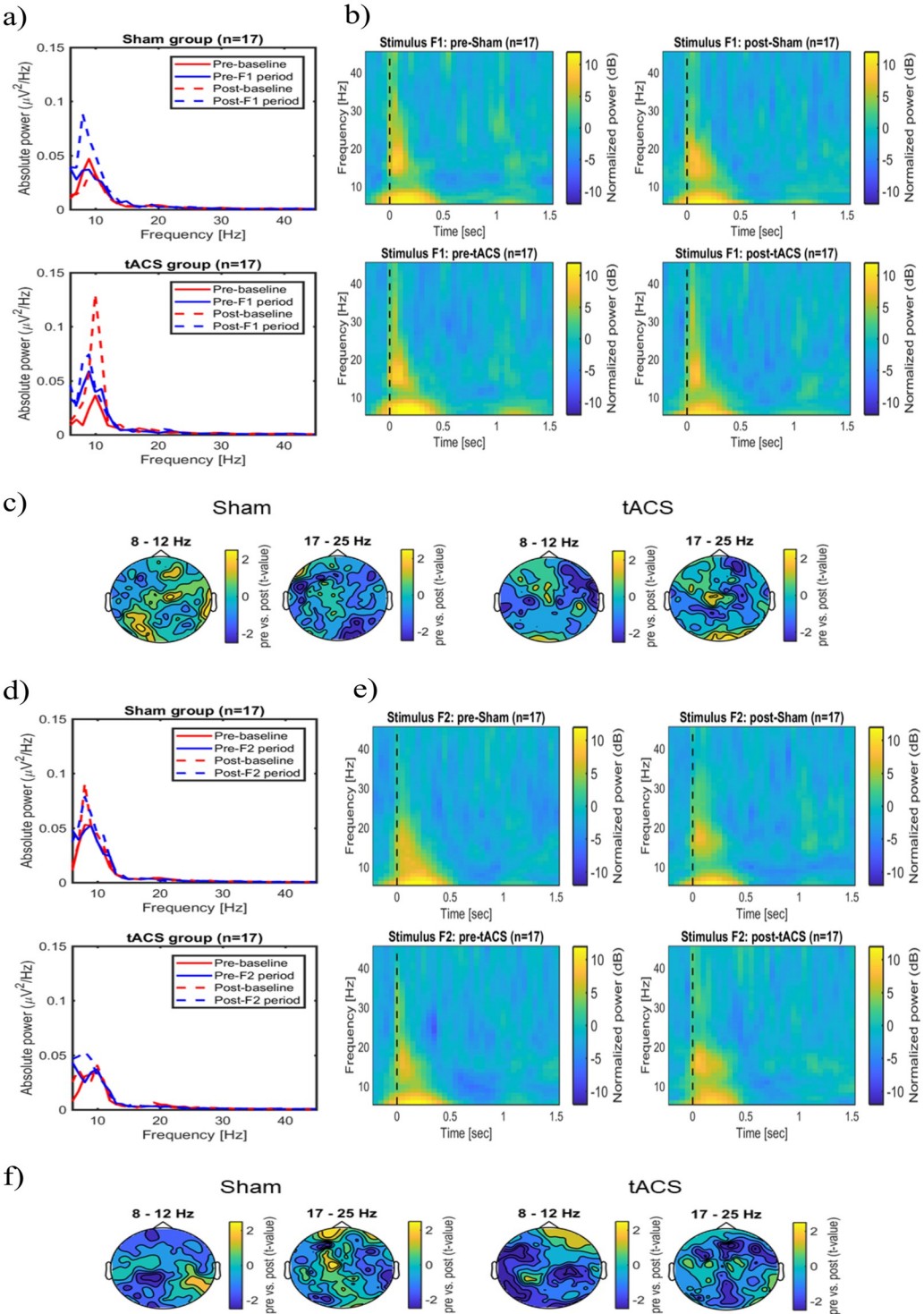

**Fig 5. Modulatory effects of tACS on the evoked oscillatory power in frequency discrimination tasks.** (a) Average absolute power changes recorded from the stimulation electrodes for pre- and post-measurements during stimulus F1. (b) Overview of the average evoked power changes during stimulus F1 presentation at the stimulation electrodes. (c) Scalp topography of the average evoked power differences before and after stimulation with Sham and tACS during stimulus F1 in a respective frequency band (Cluster-based permutation test, p > 0.05). (d) Average absolute power changes during stimulus F2 presentation recorded from the stimulation electrodes for both pre- and post-measurements. (e) Overview of the average evoked power changes during stimulus F2 presentation on the stimulation electrodes. (f) Scalp topography of the average evoked power differences before and after stimulation during F2 stimulus presentation in two frequency bands (Cluster-based permutation test, p > 0.05).

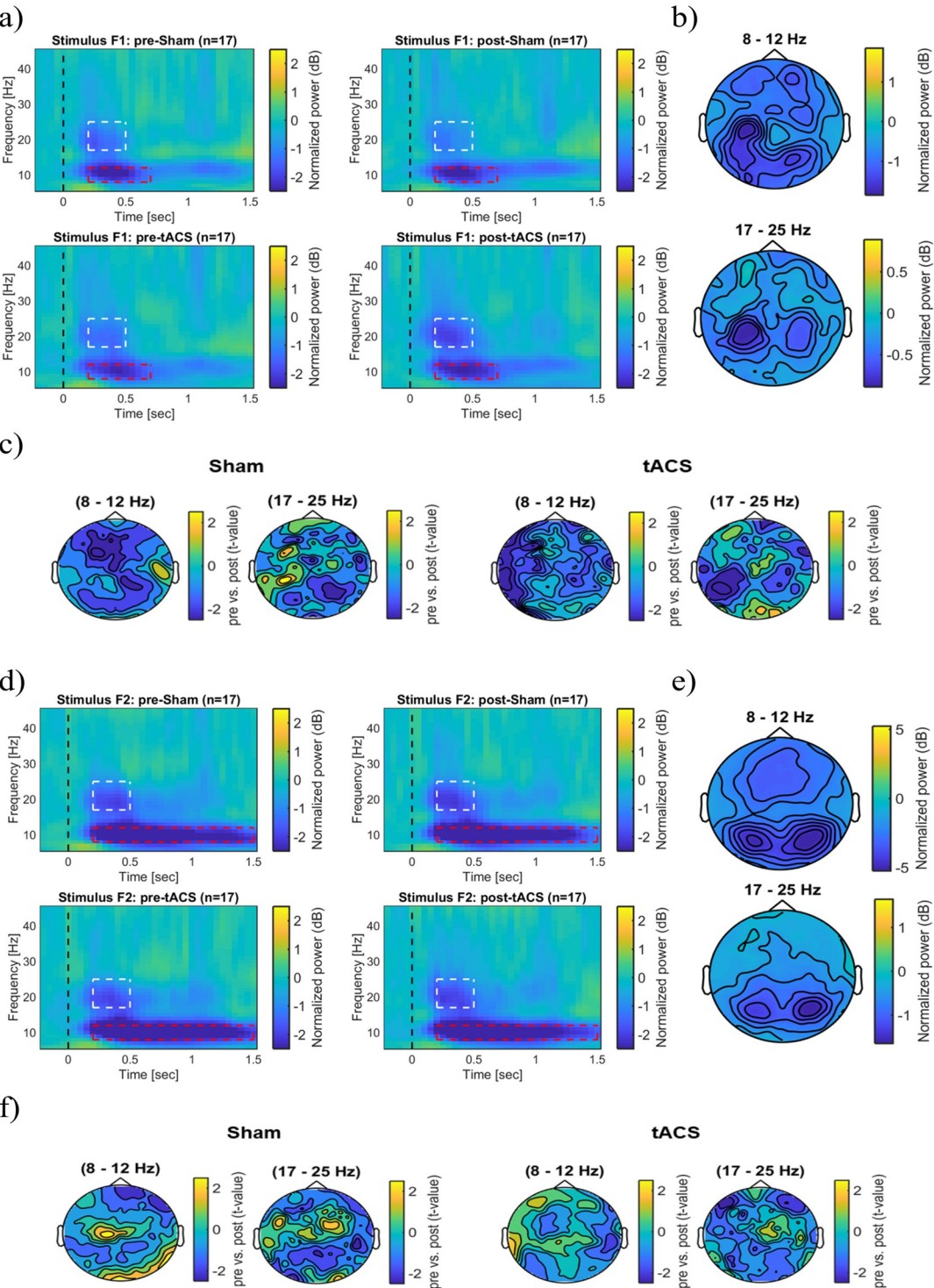

**Fig 6. Modulatory effects of tACS on induced power during and immediately after discrimination tasks.** (a) Overview of the average induced oscillatory power during and immediately after F1 stimulus presentation both before and after stimulation with Sham and tACS. Power changes were derived from the stimulation electrodes above the somatosensory cortex. (b) The figure shows the scalp topography of average induced power changes during stimulus F1. (c) Induced power differences before and after stimulation with Sham and tACS in both frequency bands during stimulus F1 (Cluster-based permutation test, p > 0.05). (d) Overview of the average induced oscillatory power during and immediately after presentation of the F2 stimulus for Sham and tACS conditions. The representations showed the average power changes on the stimulation electrodes. (e) Scalp topography demonstrates the average induced power changes during stimulus F2 at the respective frequencies. (f) Induced

power differences before and after stimulation with Sham and tACS during F2 stimulus presentation in both frequency bands (Cluster-based permutation test, p > 0.05). The dashed rectangle highlights the time-frequency window of 8–12 Hz and 17–25 Hz frequency bands.

leading to a smaller difference in N30/P30 source activity in the tACS condition (r = -0.57, p < 0.05). No relationship could be found between the angular difference and the difference of P20/N20 and N30/P30 both in sham and tACS condition at the sensor space level (see Supplementary Material, S4 Table). This indicates that an electrode vector that is parallel to the source orientation vector leads to a larger effect, i.e., larger changes in the source amplitudes. This addresses the importance of determining individualized electrode montages for optimizing the current flow to target the dipolar source orientation.

However, the angular differences did not correlate with the percentage of correct responses both in the easy and the difficult tasks after sham and tACS. There were no relationships between source activities and discrimination performance both in the easy and the difficult tasks after sham and tACS.

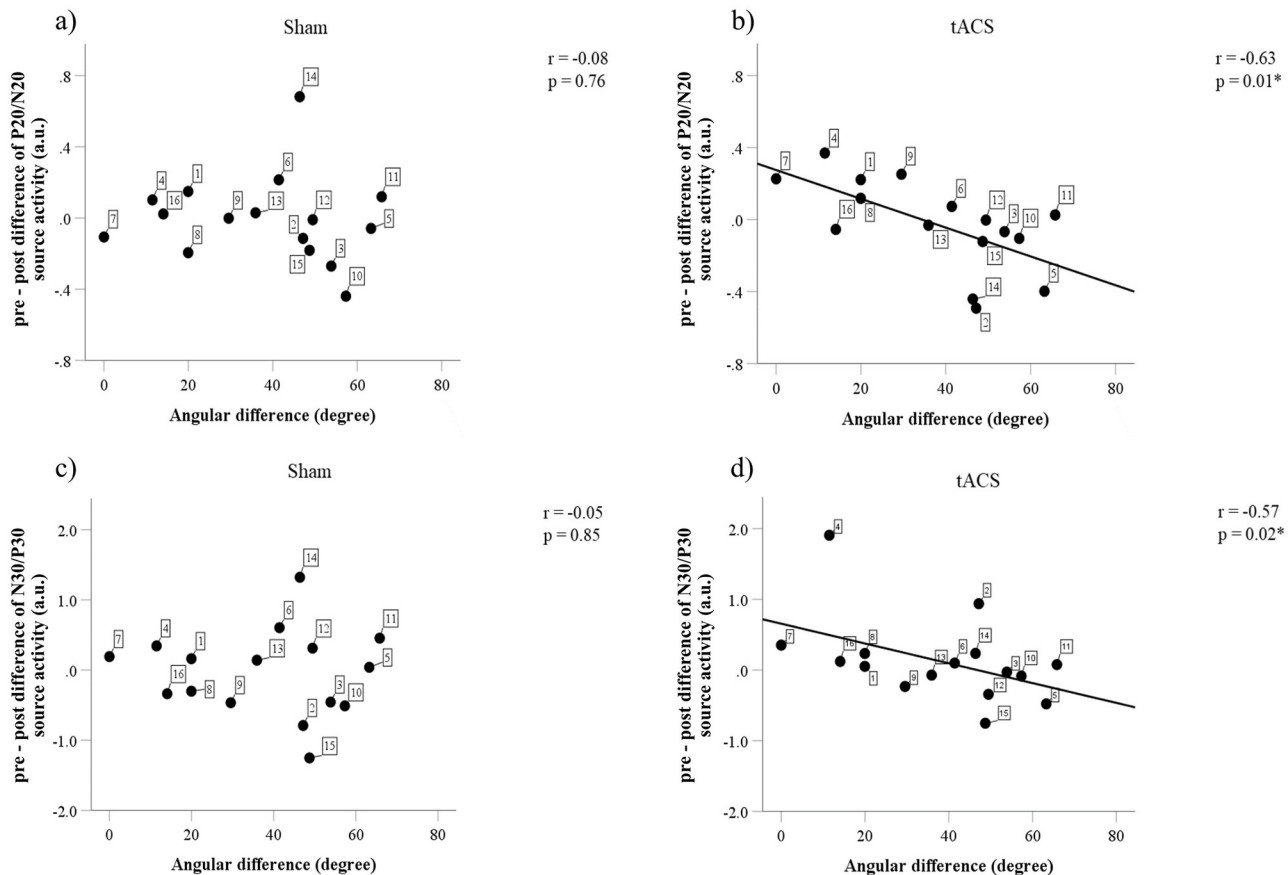

**Fig 7. Correlation between the angular difference between the stimulation electrode vector and the source orientation vector and the difference in P20/N20 and N30/P30 source activities.** (a) There is no relationship between angular difference and the P20/N20 source activity changes in the sham condition. (b) A larger angular difference is correlated with a smaller P20/N20 source activity change in the tACS condition. (c) In the sham condition, there is no correlation between the angular difference and the N30/P30 source activity changes. (d) In the tACS condition, a larger angular difference is associated with a smaller N30/P30 source activity change. The number in the filled symbol indicates a respective participant; * p < 0.05.

## Discussion

The present study examines plasticity after-effects in the somatosensory cortex induced by 140 Hz tACS and measured by the P20/N20 and N30/P30 somatosensory evoked potentials. We show that 140 Hz tACS at 1 mA at the somatosensory cortex using our current stimulation montage setup (determined by the montage using sLORETA peak thresholding) did not modulate cortical excitability on a grand average level. Also, our current 140 Hz tACS setup did not modulate the oscillatory power of alpha and beta band activities during the tactile discrimination tasks. However, importantly, we found that optimizing the individual stimulation electrode montage by targeting the individual dipolar source was associated with a clear correlation in the neurophysiological outcome of P20/N20 and N30/P30 amplitudes (Fig 7), confirming other previous findings which addressed the dependence of the induced-aftereffects on current flow directions [36–38]. Source orientation is an important factor when optimizing neuronal activation in general because it can strongly drive the induced electric field in elongated neurons [70]. This study also extends the importance of relative direction of the electric field as a significant factor from TMS-induced aftereffects in the motor cortex [71] to tES-induced aftereffects in the somatosensory cortex.

### Correlation between angular difference and magnitude change of evoked potentials and source activities

We used sLORETA in combination with the sLORETA activation peak thresholding to reconstruct a single dipole source and to optimize the position of the stimulation electrodes for targeting this dipole. However, it seems that the dipole orientation resulting from this sLORETA-thresholding procedure might be suboptimal and, in some cases, even inaccurate which in turn potentially leads to a sub-ideal or even ineffective positioning of the stimulation electrodes. A very important message for future studies is, therefore, to compare a dipole fit and sLORETA-thresholding procedures with regard to the resulting dipole orientation. The sLORETA-thresholding might also not only be done with regard to a single current density reconstruction (CDR) peak and its underlying dipole orientation, but to a larger patch of dipole sources around the sLORETA activation center. This could be either directly used for montage calculation or could be averaged into a bulk dipole orientation, always with the goal to obtain an optimal target patch to both source location and orientation. Errors or differences in source location might also translate into errors or differences in source orientation. Furthermore, the head model plays an important role in the determination of source locations and orientations, as shown in recent sensitivity analyses [72–74].

The main finding here is the inverse association between both the angular difference between the stimulation electrodes vector and the source orientation vector and the effect size of the N20. A larger angular difference led to smaller effects on the source activity. Our finding support previous results, which showed that the neuronal source orientation and position determine the direction of polarization and modulate synaptic efficacy on a cellular level [75]. Current flow direction affects the retention of learning in a ballistic movement task and modulates cortical excitability of motor cortex [37,38]. The symmetrically oscillating current flow of tACS provides the advantage, that we do only need to care about deviations from the main dipole being maximal at 90˚. With transcranial magnetic stimulation and tDCS the direction of the current flow plays a role when comparing 0˚ with 180˚. Intermittent theta burst stimulation (iTBS) after-effects are increased by 19% if the induced current flow direction matches the polarity of concurrent tDCS but is cancelled with opposite current flows [36]. In general, elongated cells such as pyramidal tract (PT-type) neurons are more direction sensitive as compared to more spherically symmetric interneurons [70].

Apart from that, our choice of the inverse method to reconstruct the P20/N20 activity by means of thresholding the sLORETA distributed source activity was based on investigations showing low localization bias in single source scenarios for this approach [76,77]. However, as mentioned above, the P20/N20 underlying source activity as seen by EEG might not fully be single dipolar due to possible additional thalamic activity for at least a small percentage of subjects, as shown by [78,79]. This might have resulted in suboptimal targeting for at least some of our subjects and thereby in suboptimal stimulation electrode montages. Indeed, we observed that not in all subjects the optimized return electrodes were overlaying the P20/N20 peak area, as expected from Helmholtz reciprocity principle [69,80]. For targeting, we found for the first time that thresholding of a current density reconstruction method like sLORETA is not appropriate for the determination of the underlying bulk target orientation. However, our choice of using the sLORETA peak thresholding inspires future investigations of the orientation-sensitivity. We recommend for future studies to use the dipole fitting method or include all dipoles of an sLORETA current density for targeting the dipolar source like the P20/N20.

## tACS effects on evoked potential amplitudes

Different protocols and stimulation parameters in the literature complicate an overview on their results. 140 Hz tACS at a current intensity of 0.7 mA (current density of 0.028 mA/cm$^2$) did not change the inhibitory circuit of the somatosensory cortex as measured by the paired-pulse depression of SEPs amplitudes after the stimulation and thus did not change perceptual discrimination performance [81]. On the other hand, Saito and colleagues showed that anodal tDCS and anodal transcranial pulsed current stimulation (tPCS) decrease the inhibitory circuit activity. A decrease in the paired-pulse depression of SEPs amplitudes resulted in an improvement in perceptual performance after transcranial random noise stimulation (tRNS) and anodal tPCS [81]. The electrode size is also an important parameter as Matsunaga and colleagues found SEP facilitation up to 60 minutes after anodal tDCS [52]. A clear discrepancy is that Matsunaga and colleagues used big patch electrodes (size 35 cm$^2$) and could not exclude a co-stimulation of motor and sensory cortex.

We cannot exclude the notion that higher tACS intensities might have produced stronger effects. Blinding higher intensities is, however, more difficult. Current intensity is a highly susceptible candidate for negative results. In tDCS studies, increasing the current intensity from 1 mA to 2 mA reversed the direction of the after-effects [82,83]. In a recent systematic dose-titration study, plasticity after-effects of anodal tDCS at 3 mA current intensity were higher compared to a standard current intensity of 1 mA [84]. Interestingly, there was also a non-linear relationship between stimulation intensities and the direction of plasticity after-effects on the motor cortex after tACS [9]. The authors showed that tACS at 0.4 mA resulted in excitability diminution, no plasticity after-effects at intensities of 0.6 mA and 0.8 mA, and excitability facilitation at 1.0 mA. All of these studies speculated that this non-linear relationship is associated with intracellular calcium increases [85]. Larger stimulation intensity increases calcium levels to induce LTP-like plasticity. Lower stimulation intensity with a lower amount of calcium increase resulted in LTD-like plasticity [82–84]. A systematic tACS dose titration study would be needed to confirm these inhibitory, excitatory and transition windows in the somatosensory cortex. An extensive review by [86] showed desirable effects of tACS at the beta frequency at intensities larger than 1 mA. In light of this evidence, we may have chosen a suboptimal stimulation intensity that falls partially in the transition zone between LTD and LTP or above the LTP zone.

## tACS effects on evoked and induced oscillatory power changes during tactile tasks

We observed rhythmic tactile stimulation-induced suppression of the mu-rhythm during a tactile task which reflects the engagement of the cortex in tactile information processing. Decreases in alpha (8–12 Hz) and beta (17–25 Hz) oscillatory power indicated activation of the somatosensory system during the tactile stimulus presentation and confirmed the inhibition hypothesis. Previous studies mentioned that suppression of alpha power (ERD) reflects activation processes, while increases in alpha power (ERS) indicate inhibitory control processes [87,88]. However, we also did not observe the modulatory effects of oscillatory activity induced by tACS as no changes in the power of alpha and beta bands occurred after stimulation. This indicates that stimulation at 'ripple' frequency did not modulate higher-order brain functions as shown by no behavioral gain. In particular, we could not reproduce the positive result of a similar tactile paradigm as shown by Pleger et al. 2006 [55]. The authors demonstrated changes in the tactile acuity after stimulation with 5 Hz repetitive transcranial magnetic stimulation (rTMS). Null effects in the behavioral performance were also demonstrated in recent studies by tACS over the somatosensory cortex at alpha, beta, and gamma frequencies [89,90]. There is a complex interaction between neurophysiological and behavioral outcome measures. For instance, the post-movement beta rebound at ~18 Hz occurred independently from rhytmical or arrhythmical rTMS motor cortex stimulation [91]. However, our finding is consistent with a previous study that showed no relationships between changes in task-related theta and gamma ERS/ERD with behavioral outcomes immediately after tRNS [16] which may support that neurophysiological measures are more sensitive than behavioral measures to evaluate the effects of tES [17].

Our method of determining individual target locations from EEG only can of course substantially be improved by combining MEG data which can stabilize source localization of the lateral and rather tangentially-oriented P20/N20 source in Brodmann area 3b clearly better than EEG [13,68,92,93]. This does not apply for radial sources such as the overlap of thalamic activity at 20 ms post-stimulus, to which the EEG is still sensitive [78,79,93,94] while the contribution of the is negligible [13,68,92,93].

## Conclusion

With the methodology we used in this study, a change in the early SEP components P20 and N30 after stimulation could only be shown by correlating the difference in source activities with the optimized direction of induced current flow. Importantly, care has to be taken in the P20 target reconstruction from EEG, not only for source location(s), but especially source orientation(s). In future studies, optimization methods based on skull-conductivity calibrated head models and if available combined EEG/MEG source analysis, where dipole fit [13,29,68] and current density reconstruction [94] results might need to be incorporated to each other to produce optimized stimulation effects.

## Supporting information

**S1 Fig. The average amplitude over subject (n = 16) of the somatosensory evoked potentials (SEPs) for each stimulation measurements.** a) The average amplitude of P20/N20. b) The average amplitude of N30/P30.
(TIF)

**S1 Table. Number of participants reporting sensations during tACS and sham stimulation.**
(PDF)

**S2 Table. Discrimination tasks performance.**
(PDF)

**S3 Table. Individual stimulation intensity in each stimulation condition.**
(PDF)

**S4 Table. Relationship between the angular differences (i.e. differences of the stimulation electrode vector and the source orientation vector) and the SEP changes both at the sensor and source space activities.**
(PDF)

**S5 Table. Relationship between the angular differences (i.e., differences of the stimulation electrode vector and the source orientation vector) and discrimination task performance.**
(PDF)

**S6 Table. Relationship between SEP source activities and discrimination task performance.**
(PDF)

## Author Contributions

**Conceptualization:** Mohd Faizal Mohd Zulkifly, Niels K. Focke, Carsten H. Wolters, Walter Paulus.

**Data curation:** Mohd Faizal Mohd Zulkifly.

**Formal analysis:** Mohd Faizal Mohd Zulkifly, Albert Lehr, Daniel van de Velden.

**Funding acquisition:** Walter Paulus.

**Investigation:** Mohd Faizal Mohd Zulkifly.

**Methodology:** Mohd Faizal Mohd Zulkifly.

**Software:** Albert Lehr, Daniel van de Velden.

**Supervision:** Carsten H. Wolters, Walter Paulus.

**Validation:** Mohd Faizal Mohd Zulkifly.

**Visualization:** Mohd Faizal Mohd Zulkifly.

**Writing – original draft:** Mohd Faizal Mohd Zulkifly.

**Writing – review & editing:** Mohd Faizal Mohd Zulkifly, Albert Lehr, Daniel van de Velden, Asad Khan, Niels K. Focke, Carsten H. Wolters, Walter Paulus.

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
