## [Decision Letter · Decision Letter 0]

18 Nov 2021

PONE-D-21-29227Directionality of the injected current targeting the P20/N20 source determines the efficacy of 140 Hz transcranial alternating current stimulation (tACS)-induced aftereffects in the somatosensory cortexPLOS ONE

Dear Dr. Mohd Zulkifly,

Thank you for submitting your manuscript to PLOS ONE. After careful consideration, we feel that it has merit but does not fully meet PLOS ONE’s publication criteria as it currently stands. Therefore, we invite you to submit a revised version of the manuscript that addresses the points raised during the review process.

We look forward to receiving your revised manuscript.

Kind regards,

Peter Schwenkreis

Academic Editor

PLOS ONE

Journal Requirements:

Reviewers' comments:

Reviewer's Responses to Questions

**Comments to the Author**

1. Is the manuscript technically sound, and do the data support the conclusions?

Reviewer #1: Yes

Reviewer #2: Partly

2. Has the statistical analysis been performed appropriately and rigorously? 

Reviewer #1: Yes

Reviewer #2: Yes

3. Have the authors made all data underlying the findings in their manuscript fully available?

Reviewer #1: No

Reviewer #2: Yes

4. Is the manuscript presented in an intelligible fashion and written in standard English?

Reviewer #1: Yes

Reviewer #2: No

5. Review Comments to the Author

Reviewer #1: The manuscript by Zulkifly describes a study on targeted transcranial electric stimulation (tES) in the sensory-motor system. In particular the authors discover that the angular match of the electric field in relation to sensory evoked activity predicts modulations of evoked activity. This is an important work that will help improving tES efficacy in my view. I quite enjoed reading this and I have only a few medium/minor comments to improve the manuscript.

comments:

1. Abstract: 'We found a decrease in the power of oscillatory mu-rhythms during and immediately after tactile discrimination tasks, indicating an engagement of the somatosensory system during stimulus encoding. On a group level the oscillatory power was neither modulated by tACS after tactile finger stimulation and nor by tACS after median nerve stimulation as shown by the lack of differences in the P20/N20 and N30/P20 amplitude compared to sham stimulation.' These two sentences don't really bring the findings accross in my opinion. In fact when reading this I got the impression that P20/N20 are cortical oscillations. Some will probably agree that ERPs and oscillations are tightly related but given that the authors analyzed oscillations separately from ERPs this seemed odd, please rephrase.

2. Methods/Participants: is the history of medication, alcohol and caffeine use available and predictive of anything? Were smokers recruited?

3. Methods: When the frequency discrimination task is described only the difference frequency is mentioned in the text, the frequencies F1/2 are only mentioned in the figure. I recommend adding this info into the text as well.

4. Methods/Main experiment: Are the digitimer stimulation intensities available somewhere? were these very different across sessions in the same subjects?

5. Methods/tACS section: Initial impedance levels are quite high with 100kOhm is this right? Is the GTEN system capable of delivering 100V (needed for 100kOhm at 1mA)? How where the impedances during the actual stimulation? Related: why is this so drastically different to the 1kOhm for the EEG channels (which use the same net)?

6. Methods/EEG: why were the data band-pass filtered in two steps and not immediately with the narrower filter?

7. Methods/analysis: for the main experiment it is stated that fieldtrip was used for analysis, though it seems it is a mix of eeglab and fieldtrip. Please correct.

8. for the spectral analysis it should be made clear that data pre-processing starts with the raw data again, otherwise one might get the impression the ERP data are further processed.

9. results/oscillatory effects: were the differences (pre vs post) statistically contrasted for sham vs tACS conditions? This might be necessary to conclude that there were no effects of the modulation.

10. Were the angular match of stimulation and ERP source correlated to behavioral modulation via tACS? It would be very interesting to know if the match of the field is predictive of a behavioral effect.

11. Figure 1: a/b are used twice in the figure. Maybe use different labelling here.

Reviewer #2: Major issue：

1. The title of this MS is not consistent with result. The efficacy of 140 tACS aftereffects were not influenced or determined by directionality.

2. The introduction is not well organized too, especially for the first two paragraph. The second paragraph of introduction is meaningless. It does not relate to the background of this research but a general point. It will be better to example why 140 Hz tACS were applied and why the discrimination tasks were used and what it the relationship between P20/N20 with such behavior performance.

3. Since the source of the P20/N20 was related to angular difference. It is interesting to re-analyze the hehavior performance data besed on the angular difference.

4. More Discussion is needed about reasons that why there is the negative correlation between source of the P20/N20 and angular difference.

Minor issue:

There are many language errors throughout the MS. It needs highly improvement. For example in the asbstract, and the sentence “They were right-handed as assessed by the Edinburgh handedness inventory [41], had no history of neurological and psychiatric illnesses and no contraindication to brain stimulation and magnetic resonance imaging (MRI)”

6. PLOS authors have the option to publish the peer review history of their article (what does this mean?). If published, this will include your full peer review and any attached files.

Reviewer #1: **Yes: **Philipp Ruhnau

Reviewer #2: No

---

## [Author Response · Author response to Decision Letter 0]

23 Feb 2022

Reviewer #1 comments: We thank and appreciate all the comments and suggestions which have allowed us to improve the manuscript further as outlined below. Please find here the responses to each comment: 

Reviewers’ comments 

1. Abstract: 'We found a decrease in the power of oscillatory mu-rhythms during and immediately after tactile discrimination tasks, indicating an engagement of the somatosensory system during stimulus encoding. On a group level the oscillatory power was neither modulated by tACS after tactile finger stimulation and nor by tACS after median nerve stimulation as shown by the lack of differences in the P20/N20 and N30/P20 amplitude compared to sham stimulation.' These two sentences don't really bring the findings across in my opinion. In fact when reading this I got the impression that P20/N20 are cortical oscillations. Some will probably agree that ERPs and oscillations are tightly related but given that the authors analyzed oscillations separately from ERPs this seemed odd, please rephrase.

Authors responses 

We rephrased the sentences and make it a clear distinction between ERPs and oscillations. Now it reads as follows:

“We found a decrease in the power of oscillatory mu-rhythms during and immediately after tactile discrimination tasks, indicating an engagement of the somatosensory system during stimulus encoding. On a group level both the oscillatory power and the evoked potential amplitudes were not modulated by tACS neither after tactile finger stimulation nor after median nerve stimulation as compared to sham stimulation”. 

2. Methods/Participants: is the history of medication, alcohol and caffeine use available and predictive of anything? Were smokers recruited? 

Authors responses:

Medication and substance use may affect the plasticity induced aftereffects. We would like to clarify that we controlled or at least documented the history of medication, alcohol, smoking and caffeine use because these might confound our studies. None of our participants consumed CNS-active medications. We documented the history of smoking and excluded those who are active smokers. Our participants were in addition advised to refrain from caffeine, smoking and alcohol use a day before the experiment and on the experimental days. 

Previous studies have shown that GABAergic drugs affect the oscillatory power of alpha and beta activity in human motor cortex [1] and CNS active drugs modulate cortical plasticity [2,3]. Alcohol use is associated with changes in brain connectivity and cortical excitability as shown by an increase in the global mean field power (GMFP) in alcohol use group [4]. Recently, we have shown that caffeine is one of the confounders in the plasticity studies and it induces different effects in caffeine naïve and caffeine-adapted subjects [5–7]. 

We included the above-mentioned information in the text and now it reads as follows: 

“They were right-handed as assessed by the Edinburgh handedness inventory [8]. Our participants had no history of neurological and psychiatric illnesses, no contraindication to brain stimulation and magnetic resonance imaging (MRI) and not the active smokers. In each session, the participant’s history of medication use, alcohol, smoking and caffeinated products use was recorded because these factors may modulate cortical excitability and plasticity, brain oscillations and connectivity [1–7]. Participants were advised to refrain from caffeine, smoking and alcohol use a day before experiment and on the experimental days”. 

3. Methods: When the frequency discrimination task is described only the difference frequency is mentioned in the text, the frequencies F1/2 are only mentioned in the figure. I recommend adding this info into the text as well.

Authors responses:

We mentioned that in the text as suggested and now it reads as follows:

“In each trial, the right index finger was stimulated with a first frequency (F1) and a second frequency (F2) for 1 sec (Fig 1b). The trial started with the subjects being asked to fixate for 1 sec a fixation cross followed by F1 after a 2 – 3 sec interval (randomly jittered in steps of 1 sec). F2 was applied after another 2 – 4 sec interval (randomly jittered in steps of 1 sec)”.

4. Methods/Main experiment: Are the digitimer stimulation intensities available somewhere? were these very different across sessions in the same subjects? 

Authors responses:

Median nerve stimulation intensity was set to the threshold at which first a visible muscle twitch can be observed. The stimulation intensity in the tACS condition was 16.41 ± 3.31 mA (mean ± SD) and in the Sham condition 14.71 ± 3.08 mA (mean ± SD), (n = 16, p > 0.05). 

Tactile stimulation intensity was set at 2.5 times of the sensory threshold. We used 2.57 ± 0.82 mA (mean ± SD) in the tACS condition and 2.65 ± 0.61 mA (mean ± SD) in the Sham condition (n =17, p > 0.05). There were slight changes in the stimulation intensities across sessions in the same subjects which justify our decision to determine the stimulation before each session (See Table 3 in the supplementary materials). 

We included this information in the result of the main text and now it reads as follows: 

1) Stimulation intensity for median nerve stimulation: 

“There was no significant difference in the median nerve stimulation intensities between tACS condition (intensity 16.41 ± 3.31 mA (mean ± SD)) and sham condition (intensity 14.71 ± 3.08 mA (mean ± SD)). (see Supplementary Material, Table S3)”. 

2) Stimulation intensity for tactile stimulation: 

“There was no significant difference in the tactile stimulation intensities between tACS condition (intensity 2.57 ± 0.82 mA (mean ± SD)) and sham condition (intensity 2.65 ± 0.61 mA (mean ± SD)). (see Supplementary Material, Table S3)”. 

5. Methods/tACS section: Initial impedance levels are quite high with 100kOhm is this right? Is the GTEN system capable of delivering 100V (needed for 100kOhm at 1mA)? How where the impedances during the actual stimulation? Related: why is this so drastically different to the 1kOhm for the EEG channels (which use the same net)? 

Authors responses:

1) Initial impedance levels are quite high with 100kOhm is this right?

We would like to clarify that the technical specifications that we stated in this manuscript follow the manual released by EGI. The GTEN 100 recording amplifier is a high input impedance amplifier that allows for higher scalp impedance without affecting the signal to noise ratio. It allows an input impedance of ≥ 1.0 GΩ (see the figures below for a detailed technical specifications). A previous study has shown that no significant attenuation in the EEG signal occurs whenever an amplifier with an input-impedance of 200 MΩ is used. Scalp-electrode impedances up to 200 kΩ still allow an accurate signal acquisition with ~0.1% error [9]. In a recent study that utilised a similar system like ours also noted that their initial impedance for stimulation were below 100 kΩ too [10].

2) Is the GTEN system capable of delivering 100V (needed for 100kOhm at 1mA)?

For each anode, the voltage budget is ± 10 volts. This yields 200µA or 50 kΩ, or 100µA at 100 kΩ. The stimulator is designed with an active loop in which the voltage is adjusted to account for varying impedance over time in order to provide a constant amperage. The system monitors the amperage and will give a high impedance warning if the circuit is not capable of providing the requested amperage with the given voltage budget. 

3) How where the impedances during the actual stimulation?

The system does not provide an opportunity to monitor the impedance during stimulation. We have access to the impedance before and immediately after the stimulation. We checked the initial impedances which were below 100 kΩ in any case before we started a stimulation. Below is the information on the mean stimulation electrode impedance (average of 12 – 20 electrodes): 

 Target Electrode Impedance (kΩ) Return Electrode Impedance (kΩ)

Before stimulation 

(mean ± S.D) 9.51 ± 5.48 11.84 ± 6.71

After stimulation

(mean ± S.D) 8.23 ± 4.15 9.34 ± 6.80

Based on the information above, this clarifies that we kept the impedance not only below 100 kΩ but substantially lower during the stimulation as well. We included this information and the text now reads as follows:

“Initial stimulation electrode impedances were kept below 100 kΩ (target electrode impedance: 9.51 ± 5.48 kΩ (mean ± S.D), return electrode impedance: 11.84 ± 6.71 kΩ (mean ± S.D))”.

4) Why is this so drastically different to the 1kOhm for the EEG channels (which use the same net)?

We are very sorry for the misinformation given in the previous manuscript version. Here we would like to confirm that the electrode impedances for the EEG channels were kept below 50 kΩ (light blue channels are acceptable impedances; see figure with the electrode impedance below). Channels having shown an impedance higher than 100 kΩ were discarded. We included this information and the text read as follows:

“The electrode impedances from all channels were kept below 50 kΩ. We applied an offline line noise filter at 50 Hz and carried out bad channel replacement using the Net Station tools (Electrical Geodesics, Inc.). Channels were labelled as bad when the system indicated a higher electrodes impedance of > 100 kΩ as a result of dried electrodes”.

6. Methods/EEG: why were the data band-pass filtered in two steps and not immediately with the narrower filter? 

Authors responses:

We followed and adapted the analysis pipeline as described by [11]. The first band-pass filter was carried out to prepare the data for independent component analysis (ICA) based artefact attenuation on continuous data. Data were low-pass filtered and high-pass filtered to improve the ICA decomposition quality. High-pass filtering between 1 and 2 Hz was reported to produce a good ICA decomposition in term of its signal to noise ratio (SNR), classification accuracy and “dipolarity” [12]. 

The second band-pass filter was conducted at the next pre-processing steps after cleaning the continuous data from stereotypical artifacts (i.e., eye blinks, lateral eye movements and electrical heartbeats) with ICA. The second filter is intended to attenuate remaining artifacts.

7. Methods/analysis: for the main experiment it is stated that fieldtrip was used for analysis, though it seems it is a mix of eeglab and fieldtrip. Please correct.

Authors responses:

We now clarify that we used the EEGLAB toolbox to do EEG data pre-processing and later we converted the data to the Fieldtrip toolbox to do further analyses. We improved our sentence formulation that now reads:

“EEG data pre-processing was carried out using EEGLAB 14.1.2 [13] as a signal processing toolbox. Later, pre-processed EEG data were analysed using the Fieldtrip [14] software with Matlab R2017a (The MathWorks, Inc., Natick, MA) as a platform”. 

8. For the spectral analysis it should be made clear that data pre-processing starts with the raw data again, otherwise one might get the impression the ERP data are further processed. 

Authors responses:

We corrected our sentence formulation to make it clear that the spectral analysis uses different data; we did the data pre-processing on the raw data. It now reads as follows:

“A continuous raw EEG data set from the electrical finger stimulation sessions was down-sampled to 500 Hz, and high-frequency noise was removed by applying a low pass FIR filter (cut-off frequency 160 Hz) and a high pass FIR filter (cut-off frequency 1 Hz) using “pop_eegfiltnew”.

9. Results/oscillatory effects: were the differences (pre vs post) statistically contrasted for sham vs tACS conditions? This might be necessary to conclude that there were no effects of the modulation. We used a cluster-based non-parametric permutation test in a Fieldtrip function to compare the SEPs amplitude changes, the oscillatory evoked and induced power changes before and after stimulation for both sham and tACS conditions (a detailed analysis description can be found in the statistical analysis section in the main text). The statistical results were reported as t-values and were shown in all figures (See Figures 4 – 6). There were no statistical differences between the SEP amplitudes, the oscillatory evoked and induced power before and after stimulation. Thus we confirmed that there was no modulation by 140 Hz tACS in the somatosensory cortex on a grand average level. 

10. Were the angular match of stimulation and ERP source correlated to behavioral modulation via tACS? It would be very interesting to know if the match of the field is predictive of a behavioral effect.

Authors responses:

There were no correlations between the angular differences and discrimination task performance after tACS (See Table 5 in the supplementary material). Also, ERP source activities changes were not correlated with discrimination task performance after tACS (See Table 6 in the supplementary material). We included these findings in the result section and discussed accordingly. It now reads as follow:

“However, the angular differences did not correlate with the percentage of correct responses both in the easy and the difficult tasks after sham and tACS. There were no relationships between source activities and discrimination performance both in the easy and the difficult tasks after sham and tACS”. 

11. Figure 1: a/b are used twice in the figure. Maybe use different labelling here.

Authors responses :

We revised the label in Figure 1 as recommended. 

Reviewer #2 comments: We thank and appreciate all the comments and suggestions which have allowed us to improve the manuscript further as outlined below. Please find here the responses to each comment:

1. The title of this MS is not consistent with result. The efficacy of 140 tACS aftereffects were not influenced or determined by directionality. 

Authors responses:

We would like to clarify that on the group level there was no efficacy. However, our main finding was on the individual level in which we showed that the direction of the dipoles (i.e. P20/N20) as measured in the angular difference between the injected current vector in the target region and the source orientation correlates with source activities. Therefore, we think the title of this manuscript is consistent with our finding. 

2. The introduction is not well organized too, especially for the first two paragraph. The second paragraph of introduction is meaningless. It does not relate to the background of this research but a general point. It will be better to example why 140 Hz tACS were applied and why the discrimination tasks were used and what it the relationship between P20/N20 with such behavior performance. 

Authors responses:

As a result of this important point addressed by the reviewer, we completely reformulated the first two paragraphs and the paragraph now reads:

“Transcranial electrical stimulation (tES) techniques allow to alter neuronal excitability by guiding low-intensity currents through the brain. Transcranial direct current stimulation (tDCS) exhibits its effects by changes in cortical excitability via alterations of neuronal resting membrane polarization, and its excitatory or inhibitory aftereffects are depending on polarity [15,16]. Transcranial alternating current stimulation (tACS) which is a non-fixed polarity protocol can also modulate cortical plasticity depending on the stimulation frequency [17–21]. A stimulation at the ‘ripple’ frequency of 140 Hz induces an excitability increase when applied at 1 mA, very similar to tDCS at 1 mA [22]. At a lower intensity of 0.4 mA an excitability decrease was demonstrated [23]. We have chosen tACS at 140 Hz and 1 mA here since it avoids the polarization seen with tDCS. Thus, we could concentrate on showing the importance of direction alignment of the injected current at the targeted area with the dipole of the primary somatosensory evoked potential (SEP) component and simultaneously excluding that the induced effects might be due to polarity differences”. 

“In this study we focus on the somatosensory cortex because of its comparatively simple dipole representations during stimulation. The P20/N20 source activity is located in Broadman area 3b which represents fingers and hand body surface [24–27]. tACS on the somatosensory cortex targeted to be aligned with the P20/N20 component might induce sensation related-effects as shown previously with tACS at alpha (10 -14 Hz), beta (16 -20 Hz) and high gamma (52 – 70 Hz) frequencies eliciting tactile sensations in the contralateral hand [28]. Also, tACS targeting the somatosensory cortex at the endogenous alpha-band activity decreases the functional connectivity of the somatosensory network [29]. Beyond neurophysiological effects, tACS also modulates behavioural outcomes such as cognitive performances [30–33] and perception [34] . We used tactile discrimination tasks to measure behaviour changes after stimulating a targeted P20/N20 source activity as its performance represents different levels of cognitive processing such as perception, recognition, working memory and decision making”. 

3. Since the source of the P20/N20 was related to angular difference. It is interesting to re-analyze the hehavior performance data based on the angular difference. 

Authors responses:

We did the analysis as suggested and found that the angular differences were not correlated with discrimination task performance in sham and tACS conditions (See Table 5 in the supplementary material) and see comment to reviewer 1. We included these findings in the result section and discussed it accordingly. It now reads as follow:

“However, the angular differences did not correlate with the percentage of correct responses both in the easy and the difficult tasks after sham and tACS. There were no relationships between source activities and discrimination performance both in the easy and the difficult tasks after sham and tACS”. 

4. More Discussion is needed about reasons that why there is the negative correlation between source of the P20/N20 and angular difference.

Authors responses: 

We wrote our discussion correspond to the related subheadings and added further information. Now the paragraph reads:

“The main finding here is the inverse association between both the angular difference between the stimulation electrodes vector and the source orientation vector and the effect size of the N20. A larger angular difference led to smaller effects on the source activity. Our finding support previous results, which showed that the neuronal source orientation and position determine the direction of polarization and modulate synaptic efficacy on a cellular level [35]. Current flow direction affects the retention of learning in a ballistic movement task and modulates cortical excitability of motor cortex [36,37]. The symmetrically oscillating current flow of tACS provides the advantage, that we do only need to care about deviations from the main dipole being maximal at 90°. With transcranial magnetic stimulation and tDCS the direction of the current flow plays a role when comparing 0° with 180°. Intermittent theta burst stimulation (iTBS) after-effects are increased by 19% if the induced current flow direction matches the polarity of concurrent tDCS but is cancelled with opposite current flows [38]. In general, elongated cells such as pyramidal tract (PT-type) neurons are more direction sensitive as compared to more spherically symmetric interneurons [39]”. 

Minor issue:

There are many language errors throughout the MS. It needs highly improvement. For example, in the abstract, and the sentence “They were right-handed as assessed by the Edinburgh handedness inventory [41], had no history of neurological and psychiatric illnesses and no contraindication to brain stimulation and magnetic resonance imaging (MRI)” 

Authors responses:

We have corrected the language, proof-read and also removed some redundancies and hope to have improved the manuscript.

---

## [Decision Letter · Decision Letter 1]

15 Mar 2022

Directionality of the injected current targeting the P20/N20 source determines the efficacy of 140 Hz transcranial alternating current stimulation (tACS)-induced aftereffects in the somatosensory cortex

PONE-D-21-29227R1

Dear Dr. Mohd Zulkifly,

We’re pleased to inform you that your manuscript has been judged scientifically suitable for publication and will be formally accepted for publication once it meets all outstanding technical requirements.

Kind regards,

Peter Schwenkreis

Academic Editor

PLOS ONE

Additional Editor Comments (optional):

Reviewers' comments:

Reviewer's Responses to Questions

**Comments to the Author**

1. If the authors have adequately addressed your comments raised in a previous round of review and you feel that this manuscript is now acceptable for publication, you may indicate that here to bypass the “Comments to the Author” section, enter your conflict of interest statement in the “Confidential to Editor” section, and submit your "Accept" recommendation.

Reviewer #1: All comments have been addressed

2. Is the manuscript technically sound, and do the data support the conclusions?

Reviewer #1: Yes

3. Has the statistical analysis been performed appropriately and rigorously? 

Reviewer #1: Yes

4. Have the authors made all data underlying the findings in their manuscript fully available?

Reviewer #1: No

5. Is the manuscript presented in an intelligible fashion and written in standard English?

Reviewer #1: Yes

6. Review Comments to the Author

Reviewer #1: I would like to thank the authors for this very thorough revision and for the detailed answers to my questions. I appreciate this.

7. PLOS authors have the option to publish the peer review history of their article (what does this mean?). If published, this will include your full peer review and any attached files.

Reviewer #1: **Yes: **Philipp Ruhnau

---

## [Editor Report · Acceptance letter]

17 Mar 2022

PONE-D-21-29227R1 

Directionality of the injected current targeting the P20/N20 source determines the efficacy of 140 Hz transcranial alternating current stimulation (tACS)-induced aftereffects in the somatosensory cortex 

Dear Dr. Mohd Zulkifly:

I'm pleased to inform you that your manuscript has been deemed suitable for publication in PLOS ONE. Congratulations! Your manuscript is now with our production department. 

Kind regards, 

on behalf of

Dr. Peter Schwenkreis 

Academic Editor

PLOS ONE